# Deep Learning-Based Auto-Segmentation of Spinal Cord Internal Structure of Diffusion Tensor Imaging in Cervical Spondylotic Myelopathy

**DOI:** 10.3390/diagnostics13050817

**Published:** 2023-02-21

**Authors:** Ningbo Fei, Guangsheng Li, Xuxiang Wang, Junpeng Li, Xiaosong Hu, Yong Hu

**Affiliations:** 1Spinal Division, Orthopedic and Traumatology Center, The Affiliated Hospital of Guangdong Medical University, Zhanjiang 524013, China; 2Department of Orthopaedics and Traumatology, The University of Hong Kong, Hong Kong SAR, China; 3Department of Orthopaedics and Traumatology, The University of Hong Kong-Shenzhen Hospital, Shenzhen 518009, China

**Keywords:** diffusion tensor imaging (DTI), image segmentation, deep learning, fractional anisotropy (FA), cervical spondylotic myelopathy (CSM)

## Abstract

Cervical spondylotic myelopathy (CSM) is a chronic disorder of the spinal cord. ROI-based features on diffusion tensor imaging (DTI) provide additional information about spinal cord status, which would benefit the diagnosis and prognosis of CSM. However, the manual extraction of the DTI-related features on multiple ROIs is time-consuming and laborious. In total, 1159 slices at cervical levels from 89 CSM patients were analyzed, and corresponding fractional anisotropy (FA) maps were calculated. Eight ROIs were drawn, covering both sides of lateral, dorsal, ventral, and gray matter. The UNet model was trained with the proposed heatmap distance loss for auto-segmentation. Mean Dice coefficients on the test dataset for dorsal, lateral, and ventral column and gray matter were 0.69, 0.67, 0.57, 0.54 on the left side and 0.68, 0.67, 0.59, 0.55 on the right side. The ROI-based mean FA value based on segmentation model strongly correlated with the value based on manual drawing. The percentages of the mean absolute error between the two values of multiple ROIs were 0.07, 0.07, 0.11, and 0.08 on the left side and 0.07, 0.1, 0.1, 0.11, and 0.07 on the right side. The proposed segmentation model has the potential to offer a more detailed spinal cord segmentation and would be beneficial for quantifying a more detailed status of the cervical spinal cord.

## 1. Introduction

Cervical spondylotic myelopathy (CSM) is characterized by chronic spinal degeneration causing structural modifications to the intervertebral discs, ligaments, etc. [1,2]. Magnetic resonance imaging (MRI) is the gold standard for diagnosing cervical spinal cord dysfunction [3]. Conventional MRIs, including T1-weighted and T2-weighted MRIs, are commonly used to obtain morphological information about the spinal cord, such as intramedullary or extramedullary abnormalities, spinal cord compression, disk herniation, etc., for identification of spinal cord injury [4,5,6,7]. Based on signal abnormalities on T2-weighted MRI, the Brain and Spinal Injury Center (BASIC) score is used to classify acute traumatic spinal cord injuries [8]. However, conventional MRI findings and clinical presentation in CSM appear disconnected, and conventional MRI cannot highlight the microstructural spinal cord abnormalities associated with CSM [2,9,10]. Diffusion tensor imaging (DTI) is a type of multi-parametric MRI and is considered a promising imaging technique for studying the ultrastructure of the spinal cord. The DTI parameter, fractional anisotropy (FA), has been used in several studies to investigate the diagnosis and prognosis of cervical spondylotic myelopathy [1,11,12]. To detect detailed neurological deficits in the spinal cord, regions of interest (ROI) can be segmented on the cord to measure the corresponding features in different tracts from the FA map [11,13,14]. A greater number of ROIs within a spinal cord leads to precise information about myelopathy. Numerous studies have demonstrated the feasibility and effectiveness of ROI-based features for providing details on the internal state of the spinal cord [11,13]. Due to its manual voxel selection and user-dependent nature, ROI delineation involves the user drawing ROIs based on perceptions of the location of underlying anatomical structures, such as the gray matter or the corticospinal tract [15]. Hand animation ROIs are prone to inter-rater and intra-rater variability and bias and are not easily transferable to large-scale studies [15]. Consequently, the manual drawing of describing ROI has serious limitations.

Some methods have been proposed for segmenting the entire spinal cord. EI Mendili et al. used T2-weighted MR images as input and employed a dual-threshold-based approach for spinal cord segmentation [16]. De Leener et al. proposed a PropSeg method based on multi-resolution propagation of tubular deformation models [17]. Chen et al. proposed a method to segment the spinal cord using deformable atlases and topological constraints that are robust to noise and artifacts [18]. In all of the above techniques, manual intervention is required, or a large database needs to be established to create the spinal cord segmentation map, and T1-weighted or T2-weighted images are the primary input. Several research studies proposed further segmenting the internal structures, gray matter (GM) and white matter (WM). Antal et al. use variational formulation to automatically detect cerebrospinal fluid, WM, and GM and combine them with shape prior to segment GM and WM [19]. Ferran proposed a two-stage pipeline using the Optimized Patch Match Label fusion (OPAL) method for segmentation of the whole spinal cord and the Similarity and Truth Estimation for Propagated Segmentations (STEPS) for the further extraction of GM and WM [20]. However, these methods can be applied to specific spinal cord levels, limiting their application to other segments. In DTI image analysis, there are some applications of segmentation methods. Marek used the semiautomatic algorithm provided by ITK-SNAP to segment the cervical spine GM and WM [21]. This method takes a T2-weighted image as input and registers the segmentation results with the DTI image. Because of the deviations between T2-weighted images and DTI images, a manual check is still necessary to guarantee the registration result. Richu applied six commonly used automatic thresholding algorithms for segmentation and vote to obtain the final segmentation results of GM and WM [14]. Although this method uses DTI images as input, the segmentation of GM and WM given by this method cannot reveal the details of each white matter column. Therefore, it should be pointed out that the existing methods of spinal cord segmentation based on DTI images provide limited ROIs within the spinal cord.

In recent years, great attention has been paid to deep learning in the medical field. One of the most widely used deep learning networks in medical image segmentation is UNet, which is based on an encoder–decoder architecture [22]. Several deep learning-based models applied UNet or UNet-like structures on spinal cord segmentation. For instance, Xiaoran et al. proposed a UNet-based, fully automatic method on 2D axial-view MRI slices for the whole spinal cord segmentation of patients with CSM [23]. Alhanouf et al. used a pre-trained MobileNet-V3 CNN model as the backbone for feature extraction, which was augmented by a set of up-sampling layers and employing skip connection similar to the UNet architecture used for spinal cord GM segmentation [24]. The deep-learning model achieved better performance on the spinal cord gray matter segmentation challenge dataset compared to Spinal Cord Toolbox (SCT), the Variational Bayesian Expectation Maximization (VBEM) method, and the Gray Matter Segmentation Based on Maximum Entropy (GSBME) method.

This paper proposes a deep learning model based on the UNet architecture for segmenting multiple ROIs within the spinal cord in DTI images. Segmentation results can be obtained simultaneously for both sides’ ventral, lateral, dorsal columns, and gray matter. Specifically, heatmap-distance loss (HDL) is proposed to train with the UNet model to make the model have better performance on the small area of column-based ROI and gray matter. This study hopes to provide a more detailed segmentation than the gray and white matter segmentation of the spinal cord and more details of the internal status of the spinal cord.

## 2. Materials and Methods

### 2.1. Study Population

An overall sample size of 89 patients with CSM was recruited for this study. All of the CSM patients are symptomatic. Clinical examination by the Japanese Orthopedic Association (JOA) produced scores of 9.78 ± 3.48, while the healthy JOA score is 17 with motor function of upper and lower limb (8 scores), sensory (6 scores) and sphincter function (3 scores). Inclusion criteria were a clinical diagnosis of CSM without a history of spinal surgery. Those with neurological disorders or prior neurological trauma were excluded. Every participant in this study completed a written informed consent form approved by the institutional review board.

DTI scanning protocol was performed with a Philips 3T Achieva scanner (Philips Medical System, Best, Netherlands). Specifically, the one-shot echoplanar imaging sequence was used. We carried out diffusion encoding in 15 nonlinear and noncoplanar directions with a b value of 600 s/mm^2^. The scanning parameters are listed as follows: field of view = 80 × 80 mm^2^, thickness of slices = 7 mm, gap between slices = 2.2 mm, fold-over direction = anteroposterior, reconstruction resolution = 0.63 × 0.63 × 7 mm^3^, and voxel resolution = 1.0 × 1.26 × 7 mm^3^, and TTE/TR = 60 ms/5 heartbeats. For the purpose of sup-pressing the fold-over effect, spatial saturation was inverted and spectral presaturation was applied. To minimize the effects of cerebrospinal fluid on cardiac vector cardiograms, cardiac vector cardiograms were triggered.

### 2.2. Manual Segmentation

The manual delineation of the ROI is an essential step for feature extraction to provide more information about the inside of the spinal cord. FA is one of the most popular parameters in DTI studies to evaluate the microstructural abnormality of the spinal cord [13,14]. Each CSM patient extracted 12 axial slices from three stacks covering the vertebrae between C2 and C7/T1. Spinal Cord Toolbox, Version 2.3 [25] was used to preprocess the DTI images, extract the B0 images, and calculate the FA map. B0 and FA images were used as input for spinal cord ROI auto-segmentation in the following experiments. The auto-segmentation ROIs and manual segmentation ROIs were superimposed on the FA images to extract the mean FA value for each ROI.

An experienced researcher in the cervical spinal cord manually drew ROIs on B0 images supplemented with FA images. The ROIs have covered more specific columns within the entire cord region, as showed in Figure 1. A total of eight regions were identified for each image to obtain detailed information about the spinal cord: lateral column (LC), dorsal column (DC), ventral column (VC), and gray matter (GM) on both sides of the spinal cord. We acquired 96 segmentation masks for each subject (8 ROIs × 12 slices). A total of 8472 segmentation masks have been created for all patients with CSM (8 ROIs × 1059 slices) except for images that are difficult to segment manually. The manual delineated ROIs were defined as the ground truth. A second expert manually segmented these ROIs to enhance confidence in identifying these eight ROIs. Whenever there was a discrepancy, a third expert was consulted until a consensus was reached.

### 2.3. Segmentation Models

The UNet [22] was introduced for its popularity in medical image segmentation. The model structure is similar to a fully convolutional network, which consists of two parts. The first part consists of layers of convolution and max pooling (known as the encoder). B0 images will be used as input to extract multi-scale features. The second part is mirror-symmetric to the first part (also known as the decoder), consisting of transposed convolution and up-sampling layers for feature expansion of a feature vector of an image of sizes corresponding to the original medical image. Several skip connections between the encoder and decoder were added to further utilize the multi-scale feature of the encoder for feature expansion. Figure 2 demonstrates the detailed parameters of the UNet model.

Generally, the feature extracted from the encoder highly affects the segmentation performance. Two different depth general network structures, VGG-16 [26] and ResNet50 [27], were used as the backbone for feature extraction. Two segmentation tasks corresponded to two models with different backbones for feature extraction, including vgg_9classes for nine classes (plus background) segmentation using VGG16 as the backbone and resnet_9classes for nine classes (plus background) segmentation using ResNet50 as the backbone.

A challenge that needs to be noticed in this study is the small object of manual segmentation. In our dataset, all ROIs are small areas, and the largest ROI only takes about 1/1000 of the whole DTI images from Figure 3. The challenge for model training comes from small-shape annotations that contribute less to the loss than background annotations, which makes the model prone to classify a pixel as background because image segmentation is a pixel-wise classification. Wentao et al. [28] use the Dice loss combined with the focal loss to segment small-volume organs in the Organs-At-Risk delineation problem and achieve good segmentation accuracy, while the Dice loss learns the imbalanced class distribution, and the focal loss trains the model to learn hard segmented annotations for these small-volume organs. Hence, this hybrid loss was selected for the training of two models. The total loss can be formulated as follows:(1)$$TP\left(i\right)=\overset{N}{\underset{n=1}{\sum }}p_{n}\left(i\right)g_{n}\left(i\right)$$
(2)$$FN\left(i\right)=\overset{N}{\underset{n=1}{\sum }}\left(1-p_{n}\left(i\right)\right)g_{n}\left(i\right)$$
(3)$$FP\left(i\right)=\overset{N}{\underset{n=1}{\sum }}p_{n}\left(i\right)\left(1-g_{n}\left(i\right)\right)$$
(4)$$L_{total}=L_{DICE}+L_{Focal}=C-\overset{C-1}{\underset{i=1}{\sum }}\frac{TP\left(i\right)}{TP\left(i\right)+FN\left(i\right)+FP\left(i\right)}-\frac{1}{N}\overset{C-1}{\underset{c=1}{\sum }}\overset{N}{\underset{n=1}{\sum }}g_{n}\left(i\right){\left(1-p_{n}\left(i\right)\right)}^{2}\log\left(p_{n}\left(i\right)\right)$$
where *TP*(*i*), *FN*(*i*), and *FP*(*i*) are the true positives rate, false negatives rate, and false positives rate for the classes. $p_{n}\left(i\right)$ is the predicted output value for the pixel *n* class *i*, $g_{n}\left(i\right)$ is the ground truth label for pixel *n* class *i*, and *C* is the total number of anatomies included in the background. *N* is the total number of pixels in one slice of a B0 slice image.

However, the performance of the segmentation model using B0 images as input on the test dataset was not satisfactory. Although we tried to use the FA images as input, the segmentation performance was also unsatisfactory. Except for the small object, there is another challenge in this dataset: the difference in pixel values between different classes is minimal, making it difficult to distinguish the boundaries of different ROIs. Based on the above challenges and performance on the test dataset, we believe the Dice loss combined with focal loss is unsuitable for this research.

Inspired by the landmark localization based on two-dimensional (2D) heatmaps [29] and the boundary-weighted map from Qikui’s research [30], we proposed the heatmap-distance loss (HDL). The loss can be formulated as follows:(5)$$HDL=\overset{9}{\underset{i=1}{\sum }}\|H_{i}\left(X\right)-\hat{H_{i}}{\left(X)\right\|}_{2}$$
(6)$$\hat{H_{i}}\left(x\right)=\{\begin{array}{cc} 1, & \;\;x\in X_{landmark}\\ \exp\left(-\frac{\|\max{\left(x-X_{landmark})\right\|}_{2}^{2}}{2\sigma^{2}}\right), & x\notin X_{landmark} \end{array}$$
where *X* denotes the set of pixels in 2D space from any class ground truth mask, and *x* represents any pixels from *X*. Let $X_{landmark}$ denote the set of landmark points whose pixel value is equal to one. The notation ‖⋅‖2 denotes the L2 or Euclidean distance. $\hat{H_{i}}\left(x\right)$ refers to the ground truth heatmap for the *i*th class of ground truth masks, while $H_{i}\left(X\right)$ is the predicted heatmap for the *i*th class. The pixel value of the heatmap $\hat{H_{i}}\left(x\right)$ is created based on formulation (6). When the point from *X* has the same coordinate with the point in $X_{landmark}$, the value of $\hat{H_{i}}\left(x\right)$ is equal to one. Otherwise, each point will calculate the L2 or Euclidean distance between the point and each landmark point. The max distance between each pixel and the set of landmark points will be used as input to a Gaussian function with variance $\sigma^{2}$ to generate the value of $\hat{H_{i}}\left(x\right)$. The rationale behind the design is that the small area of segmentation can be expanded to a large area, and the pixels created by Gaussian based on distance could make the small area contribute to the training process. Based on the test dataset performance and the model structure’s complexity, a better encoder between VGG16 and ResNet50 would be used as a backbone trained with the proposed loss.

### 2.4. Training and Evaluation

We use 81% (857 images) of our dataset for training, 9% (96 images) of our dataset for validation, and 10% (106 images) as a test dataset. The UNet model is trained on the training dataset and validated on the validation dataset. The final performance is exclusively reported on the test set.

The experiments were performed in the Pytorch environment, and we trained the segmentation model on NVIDIA RTX3080Ti. The batch size was set to 16 based on the memory of the GPU and computer. We used the Adam optimizer with a momentum of 0.9 and a cosine learning rate strategy with a learning rate range from 1 × 10^−4^ to 1 × 10^−6^; the number of epochs is 500. The Dice loss combined with focal loss and the HDL loss was trained under the same hyperparameters. The threshold definition is an essential step that transforms the model output into segmentation results. The threshold values for different segmentation classes are selected from 0 to 1 with intervals of 0.001 on validation datasets, and the threshold value of the best performance was to evaluate the final performance on the test dataset.

For evaluation of the proposed method, three metrics for each class are used, including the Recall (also known as True Positive Rate), which represents a method’s ability to segment pixels as a proportion of all correctly labeled pixels, the Precision, which measures the ratio of correctly segmented pixels and the Dice coefficient, which is a similarity index between two masks. The following equations define these three metrics: GT refers to ground truth, and PM refers to predict mask. Symbols FN, FP, and TP are explained as follows: true positive (TP) if it was a pixel in GT mask and it was segmented in PM; false positive (FP) if it was not a pixel in GT mask and it was segmented in PM; and finally, false negative (FN) if it was a pixel in GT mask and it was not segmented in PM:(7)$$DICE=\frac{2\left|GT\cap^{\;}PM\right|}{\left|GT\right|+\left|PM\right|}$$
(8)$$Recall=\frac{TP}{TP+FN}\;$$
(9)$$Precision=\frac{TP}{TP+FP}$$

To further evaluate the accuracy of the mean FA value from the segmentation model, a metric $FA_{error}^{i}$, the percentage of absolute error of FA was utilized to compare the mean value of the ground truth mask superimposing on the DTI metric map (FA map) and the prediction segmentation mask superimposing on the DTI metric map (FA map), which is formulated as follows:(10)$$FA_{error}^{i}=\frac{\left|FA_{GT}^{i}-FA_{PM}^{i}\right|}{FA_{GT}^{i}}$$
where $FA_{GT}^{i}$ refers to the FA mean value from the ground truth mask of the *i*th segmentation class and $FA_{PM}^{i}$ refers to the FA mean value from the predicted mask of the *i*th segmentation class.

The performance of the algorithm was evaluated using basic statistics, such as the mean and standard deviation of $FA_{error}^{i}$. Outliers were defined as slices of DTI images that did not obtain the segmentation results from the model. After removing these outliers, the performance was calculated. The author performed an outlier assessment by calculating the percentage of outliers within this dataset.

## 3. Results

### 3.1. Encoder Comparison

Figure 4 demonstrates the performance of the UNet with different backbones on the test dataset (B0 as input images). We used the VGG16 and ResNet50 as the encoder for UNet training with a hybrid loss of Dice loss and focal loss. The UNet with VGG16 as the backbone achieves the highest Dice value of 0.65 on the right dorsal column and reaches the lowest Dice value of 0.48 on the left ventral column. Figure 4 demonstrates that the UNet model with VGG16 as the backbone performs better than using ResNet50 as the backbone on all segmentation classes. Therefore, VGG16 will be used as a backbone for UNet trained with the proposed loss function.

### 3.2. Model Performance

Figure 5 shows the segmentation result of several samples using the proposed model. In these pictures, the line in blue color is by GT and the line in red color is by PM.

Figure 6 demonstrates the performance on test datasets (FA as input images) of UNet training with hybrid loss and HDL loss. The UNet with VGG16 as the backbone achieves the highest Dice value of 0.69 on the left dorsal column and the lowest Dice value of 0.54 on the left gray matter. Figure 6 demonstrates that the UNet model trained with the proposed HDL loss performs better than the hybrid loss on all segmentation classes. Table 1 presents the overall performance of UNet using the proposed HDL loss.

To eliminate the effect of different kinds of images, Figure 7 demonstrates the performance of the UNet model training under different loss functions and using different types of DTI images. UNet model training with the proposed HDL loss using FA as input images achieves the highest performance.

There is a situation in which several images could not obtain the segmentation result from the model defined as the outlier slices. Table 2 illustrates the statistics of performance removing these outlier slices. The Dice coefficients of the left lateral column, ventral column, and gray matter were improved by 1%, 4%, and 3%, respectively. The Dice coefficients of the right lateral column, ventral column, and gray matter were improved by 2%, 4%, and 6%, respectively. The “Outliers” column demonstrated the percentage of DTI slices that cannot obtain output from segmentation.

### 3.3. Accuracy of FA Prediction

Furthermore, segmentation aims to extract the mean FA value within ROIs from the spinal cord. Therefore, we calculate the mean FA values from auto-segmented and manual-segmented ROIs. A series of higher intra-class correlation coefficients were found in Figure 8, indicating excellent agreement between the FA from segmented ROIs and the FA from ground truth ROIs. Table 3 demonstrates the percentage of the absolute error between the mean FA value from the ground truth mask and the mean FA value from the prediction mask. The percentage of the mean absolute error for lateral, dorsal, ventral, and gray matter was 0.07, 0.07, 0.11 and 0.08 on the left side as well as 0.07, 0.1, 0.1 and 0.07 on the right side, respectively.

## 4. Discussion

Microstructural spinal cord abnormalities are vital for CSM patients. DTI can reveal the microstructural change in the spinal cord compared to conventional MRI. However, the DTI analysis based on the whole cord region cannot reveal the details of the spinal cord. The subtle abnormality of DTI parameters caused by minor microstructure impairment in a small ROI may be missed due to the whole cord analysis dilution effect. ROI-based DTI analysis could avoid the drawback and enable the quantitative evaluation of specific regions with the spinal cord. This study combines UNet with the proposed heatmap distance loss to segment multiple ROIs within the spinal cord on DTI-related images from CSM patients. Many previous studies in spinal cord segmentation mainly focus on T1-weighted [31,32,33] and T2-weighted images [23,34,35] instead of DTI-related images. The related studies in spinal cord segmentation approaches on DTI images are semi-automatic [14,21] or diffusion-tensor-tracking-based [36] instead of fully automatic. The segmentation region of associated studies is the whole cord [23,31,32,33,35,36,37] or gray matter/white matter [14,19,20,24,38,39,40,41] instead of multiple ROIs within the spinal cord. To the authors’ knowledge, this is the first study using the deep learning method to fully automate and segment the multiple ROIs within the spinal cord on DTI images from CSM patients. We focused on developing an approach to help clinicians provide a more detailed description of spinal status of CSM patients by providing ROI-based features to facilitate ROI-specific analysis of DTI images.

We used two backbone networks to extract the feature within DTI-related images. VGG16 [26] and ResNet50 [27] are well-known convolutional neural networks in many natural-images-related tasks, and these networks achieve good performance on feature extraction. We hypothesize that the higher-level feature would be beneficial for segmentation. Figure 4 illustrates that the hypothesis is wrong. Figure 3 demonstrates that the percentages of pixels for all classes are small than 0.0011. Hence, the advanced features would not be suitable for this task because the convolutional process from a deeper layer would eliminate the efficient information for the training of the segmentation model.

The obstacle to fulfilling the segmentation in this dataset is the ROIs within the spinal cord. Figure 5 demonstrates several samples with the auto-segmentation result. Because the spinal medulla is misshaped in CSM patients, the ROIs contain variable sizes. To deal with the above issue, several loss functions have been proposed. The Dice loss and focal loss are commonly used for segmentation. Wentao [28] used focal loss combined with Dice loss for organ-at-risk (OAR) segmentation, especially for small-volume organ segmentation, achieving better segmentation accuracy. In this research, we use the focal loss combined with Dice loss to train the UNet to fulfill the segmentation of ROIs within the spinal cord. Figure 4 demonstrates that the segmentation performance of sub-ROIs with the spinal cord is not satisfied. The possible reason is that the region of sub-ROIs from the spinal cord is smaller than the volume of OARs in Wentao’s research, and the boundary of these ROIs in DTI-related images is not as clear as in computed tomography (CT) images. We proposed a new heatmap distance loss function inspired by landmark localization. Figure 6 demonstrates that the UNet with the proposed HDL achieves the highest segmentation performance for all classes. The basic principle of HDL is that we manually expand the ground truth area by designing the pseudo-probability, which makes the small area significantly contribute to the model training.

However, the higher performance in Figure 6 was based on the FA images, while the lower performance in Figure 6 was based on the B0 images. Regardless of the image types, Figure 7 illustrates that the UNet model training using HDL performs better than hybrid loss. The B0 images have a wide range, as well as the CT image in Wentao’s research, while the value range of FA images is from 0 to 1. A wide range of values would be beneficial for the convolutional process to extract the high-level feature and the hybrid loss, which could be the reason that VGG_B0_hybird_loss has better performance than VGG_FA_hybird_loss. Figure 7 also illustrates that the pixel value of images has less effect on the model training using proposed HDL loss.

Figure 8 demonstrates the correlation between the mean FA of auto-segmented ROIs and the mean FA of ground truth. The high intra-class correlation on the diagonal position of the correlation matrix demonstrated the high agreement between the mean FA value from auto-segmented ROIs and ground truth ROIs. Several values on the correlation matrix need to be explored. It should be noted that the correlations from several pairs are high, including {L_DC, R_DC}, {R_DC, L_DC}, {L_VC, R_VC}, {R_VC, L_VC}, {L_GM, R_GM}, and {R_GM, L_GM}. This result illustrated that the ROI of Pred_L_DC or Pred_R_VC intersects with GT_L_DC or GT_R_VC, and there is a high agreement between the left and right of DC, VC, and GM, which could be the factor that affects the segmentation performance. Table 3 demonstrates the absolute error percentage between the FA of auto-segmented ROIs and ground truth ROIs on the test dataset. The mean distance between the mean FA value obtained from the segmentation model and the mean FA value obtained from manually segmented ROIs is small, demonstrating that the mean FA value extracted from the segmentation model has the potential for detailed spinal cord status evaluation, such as the mean FA value extracted from the ground truth. The spinal medulla misshaped in CSM patients still significantly affects segmentation performance. However, most of the segmentation result is close to the manual segmentation, and most of the auto-segmentation results are on the related region, which could be why the distance between mean FA values from the auto-segmentation and mean FA values from manual segmentation are small.

Clinically, DTI has three applications. Firstly, it has the ability to identify spinal cord structures and injuries, localization, and diagnosis of CSM. Cui et al. [10]’s research demonstrated that the orientation entropy from DTI analysis is a valuable tool for identifying the pathological level in multilevel CSM patients. Shu-Qiang Wang [42] combined the eigenvalues from DTI and machine learning methods to achieve satisfactory performance in recognizing the levels of CSM. According to Monika Skotarczak [43], the DTI metrics can be used as a biomarker to illustrate the microstructural disorder of the spinal cord not visible on conventional MRI. Secondly, there is the indication of surgery. As demonstrated by Karsten Schöller et al. [44], DTI metrics achieve higher sensitivity in identifying levels requiring decompression surgery than increased MRI signals. Thirdly, there is dynamic spinal function examination and prognosis assessment. For instance, Ellingson et al. [45]’s study demonstrated that DTI has the potential to monitor symptomatic patients and asymptomatic patients, and Chun Yi Wen et al. [46]’s research revealed that FA is a biomarker for surgical outcomes. The auto-segmentation of ROIs with the spinal cord DTI has the potential to reduce inter-rater and intra-rater variability in manually drawn ROIs and help the entire DTI analysis workflow be automated.

There are several limitations. Firstly, there is space for improvement in segmentation performance. Based on the findings from Figure 5, the deep learning model with fewer layers will be explored, which will be regarded as part of the future extension of this research. Secondly, the intersection between the ROI of Pred_L_DC and the ROI of GT_R_DC, the ROI of Pred_R_DC and the ROI of GT_L_DC, the ROI of Pred_L_VC and the ROI of GT_R_VC or the ROI of Pred_R_VC and the ROI of GT_L_VC is a problem that could be regarded as one reason to improve the segmentation performance of DC(L & R) and VC(L&R). A specified preprocessing or postprocessing method needs to be explored in the future. Thirdly, the diffusion tensor for calculation of the DTI metric is reconstructed based on a series of diffusion-weighted images and the b-matrix that integrates the parameters of diffusion-sensitizing gradients. However, many factors influence the accuracy or reliability of the diffusion tensor evaluation, including image noise [47], eddy currents [48], diffusion gradient nonlinearity [47], and others. For example, in the circumstance of low anisotropy, imaging noise could lead to the wrong estimation of eigenvalues, therefore causing an overestimation of DTI metrics (such as FA) due to the incorrect order of eigenvalues [49], and diffusion gradient inhomogeneities would cause distortion of the diffusion tensor’s eigenvalues as well as rotation of the eigenvectors [50]. The diffusion gradient inhomogeneity is an important source of systematic error, and several methods have been proposed to correct the error due to diffusion gradient inhomogeneity [51,52,53,54]. They are either based on a calibration that uses an anisotropic phantom as a reference for estimation of the diffusion tensor, mapping the actual magnetic field, or using the coil system’s manufacturer-provided specifications. The b-matrix spatial distribution in DTI (BSI-DTI) [55] is a frequently employed technique for correcting this kind of inaccuracy and has the capacity to minimize the impact on the assessment of DTI metrics. To reduce the effects of inhomogeneous magnetic field gradients and make the data as accurate as possible, we intend to adopt the BSI-DTI approach in the future. Finally, different types of input images potentially affect the performance of segmentation, and many factors of MRI sequences can affect the DTI images. The MRI scanner involves the measurement of DTI metrics (such as FA) [56]. Except for the MRI scanner, several studies demonstrated that the parameters of MRI sequence, such as the b-value [57], echo time [58], the number of DTI directions [59], and the number of signal acquisitions [60], could influence the diffusion quantification. In order to expand the scope of application of the model, it is necessary to extend the diversity of the DTI image dataset: for instance, acquiring the DTI images of CSM patients with different b-value and different numbers of DTI directions.

In future work, we will consider the continuation path for this research in the following steps. Firstly, we will continue to expand the size of our dataset, as the sample size in this research is insufficient for deep learning training. Secondly, weak-supervised training will be used on the new dataset. A new and helpful segmentation model will be developed using the manual segmentation result in this research. Thirdly, DTI images of CSM patients will be classified into several categories based on the types of CSM compression or the classification result from unsupervised learning. The segmentation model will be trained on the specified cases to verify the algorithm’s effectiveness in the specific category of CSM cases.

## 5. Conclusions

This paper proposed using UNet and the heatmap distance loss to automatically segment the sub-ROIs within the spinal cord on DTI images from CSM patients. The performance of the segmentation model and the agreement between the mean FA value of auto-segmented ROIs and the mean FA value of manually segmented ROIs demonstrates that the deep learning model has the potential to provide more details of the internal status of the spinal cord.

## Figures and Tables

**Figure 1 diagnostics-13-00817-f001:**
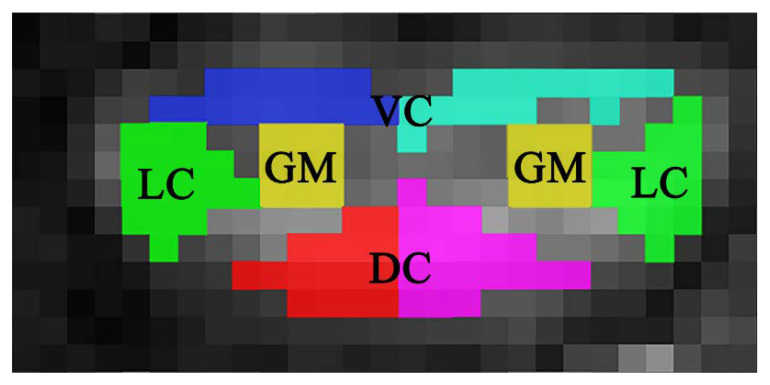
ROIs covering the whole spinal cord.

**Figure 2 diagnostics-13-00817-f002:**
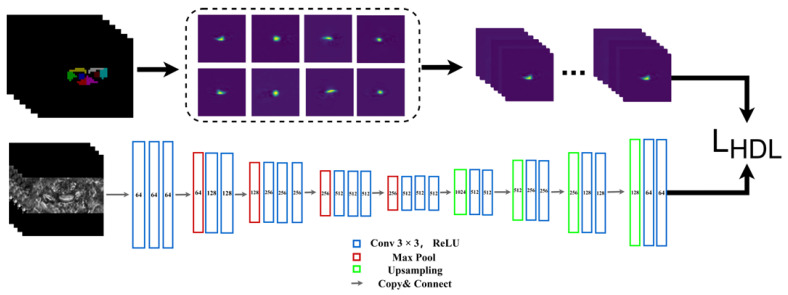
Structure of the segmentation network.

**Figure 3 diagnostics-13-00817-f003:**
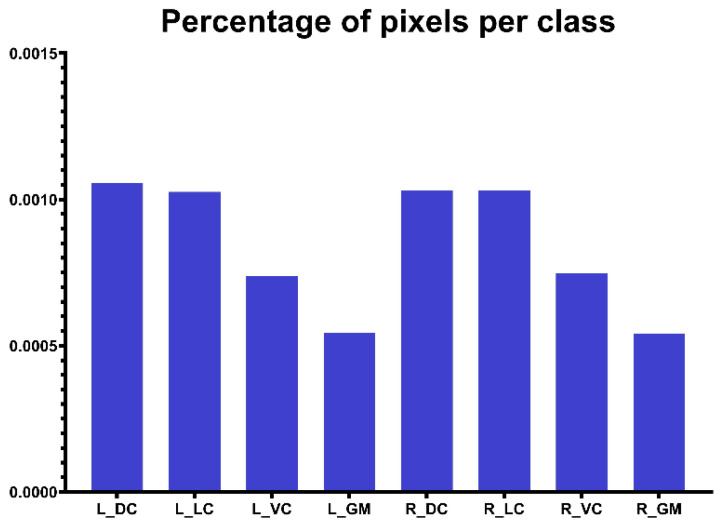
The percentage of pixels for each class on DTI images.

**Figure 4 diagnostics-13-00817-f004:**
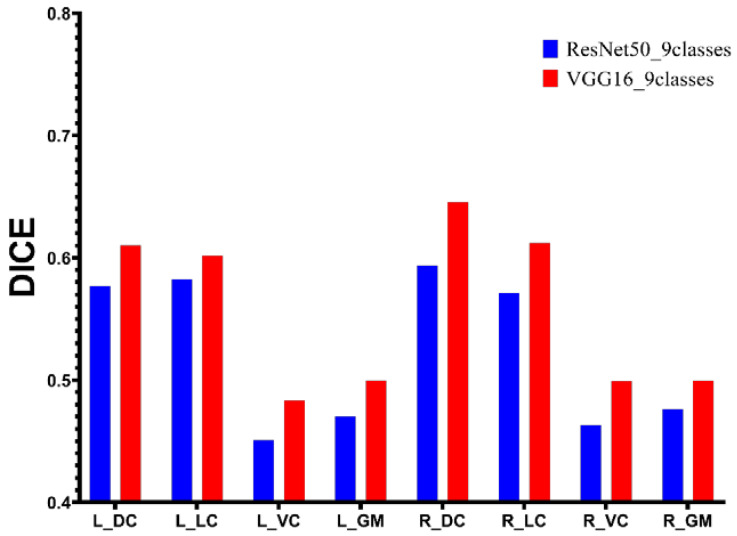
The comparison between different encoders.

**Figure 5 diagnostics-13-00817-f005:**
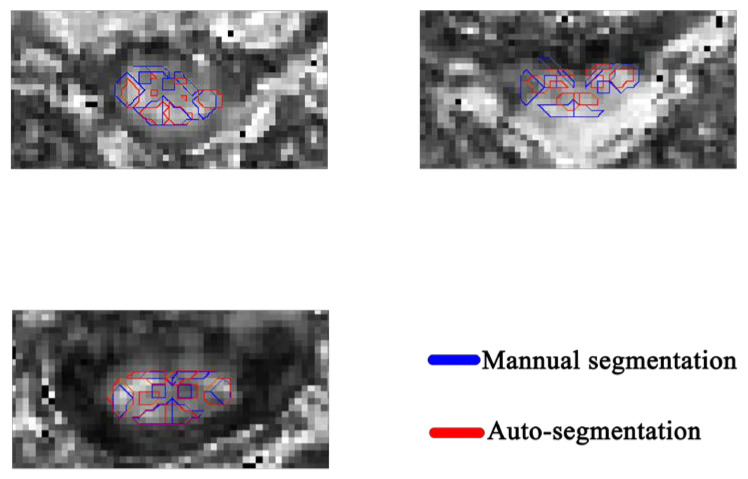
Several segmentation results with the proposed model.

**Figure 6 diagnostics-13-00817-f006:**
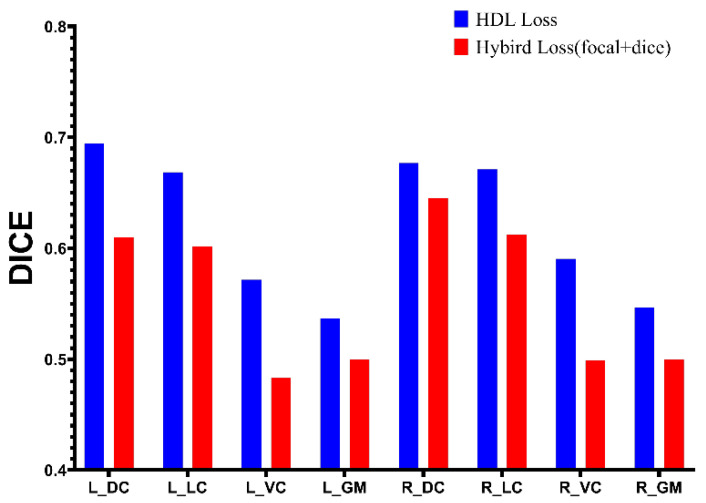
The performance comparison of UNet with different losses.

**Figure 7 diagnostics-13-00817-f007:**
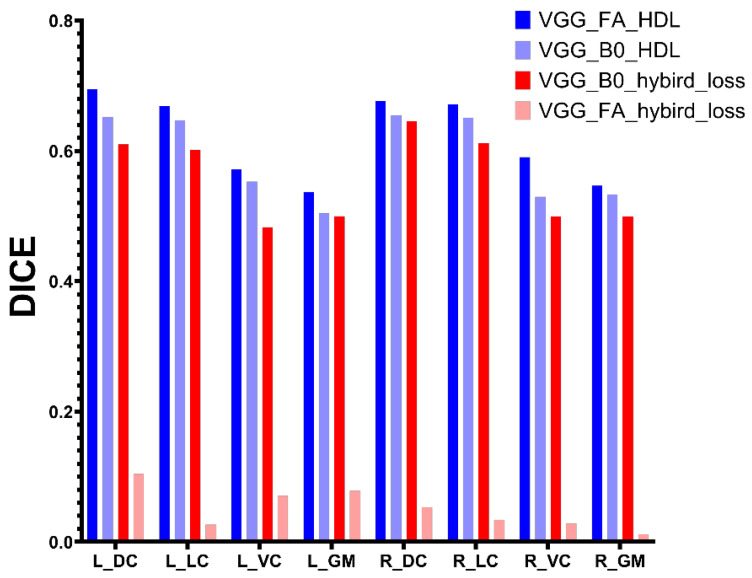
The performance comparison of UNet with different losses using different DTI images.

**Figure 8 diagnostics-13-00817-f008:**
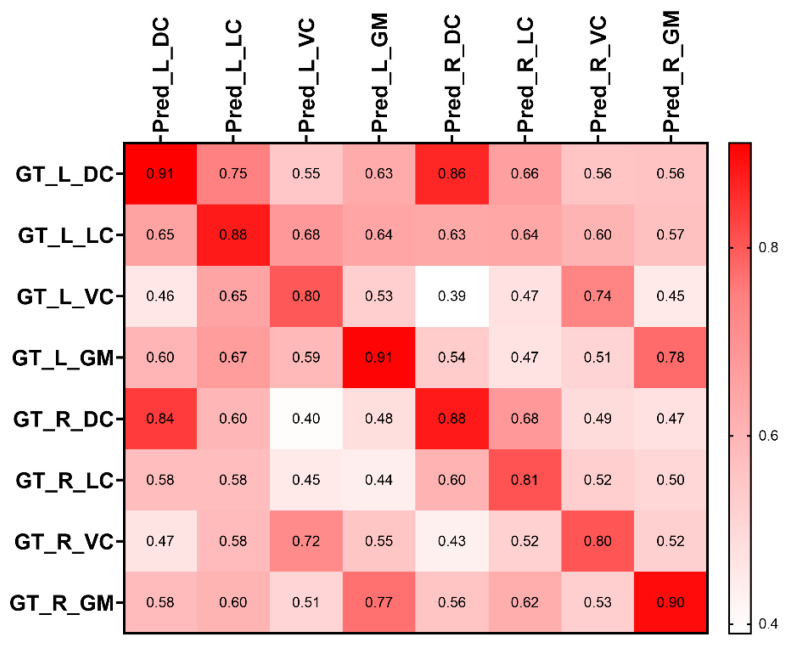
The correlation between FA of predicted segmentation ROIs and FA of ground truth ROIs. Pred_: the ROIs from the proposed segmentation model; GT_: the ROIs from the manual segmentation.

**Table 1 diagnostics-13-00817-t001:** The performance of UNet with proposed HDL loss.

Class	Dice	Recall	Precision
Left dorsal column	0.69 ± 0.19	0.78 ± 0.23	0.64 ± 0.2
Left lateral column	0.67 ± 0.24	0.74 ± 0.29	0.63 ± 0.23
Left ventral column	0.57 ± 0.29	0.62 ± 0.33	0.55 ± 0.28
Left gray matter	0.54 ± 0.28	0.59 ± 0.33	0.53 ± 0.28
Right dorsal column	0.68 ± 0.21	0.74 ± 0.25	0.65 ± 0.2
Right lateral column	0.67 ± 0.22	0.74 ± 0.26	0.65 ± 0.22
Right ventral column	0.59 ± 0.26	0.63 ± 0.29	0.58 ± 0.27
Right gray matter	0.55 ± 0.31	0.56 ± 0.34	0.57 ± 0.32

**Table 2 diagnostics-13-00817-t002:** The performance of UNet with proposed loss removed outlier slices.

Class	Dice	Recall	Precision	Outliers
Left dorsal column	0.69 ± 0.19	0.78 ± 0.23	0.64 ± 0.2	0
Left lateral column	0.68 ± 0.23	0.76 ± 0.27	0.64 ± 0.21	2%
Left ventral column	0.61 ± 0.25	0.67 ± 0.3	0.59 ± 0.25	7%
Left gray matter	0.57 ± 0.25	0.63 ± 0.3	0.56 ± 0.25	7%
Right dorsal column	0.68 ± 0.2	0.74 ± 0.24	0.66 ± 0.2	1%
Right lateral column	0.69 ± 0.18	0.76 ± 0.23	0.67 ± 0.19	3%
Right ventral column	0.63 ± 0.23	0.67 ± 0.26	0.61 ± 0.24	6%
Right gray matter	0.61 ± 0.27	0.63 ± 0.3	0.63 ± 0.27	10%

**Table 3 diagnostics-13-00817-t003:** The percentage of absolute error between two types of FA value.

Class	$FA_{error}^{i}$
Left dorsal column	0.07 ± 0.07
Left lateral column	0.07 ± 0.11
Left ventral column	0.11 ± 0.17
Left gray matter	0.08 ± 0.14
Right dorsal column	0.07 ± 0.08
Right lateral column	0.1 ± 0.11
Right ventral column	0.1 ± 0.23
Right gray matter	0.07 ± 0.09

## Data Availability

Because the limits of ethic approval, the original DTI dataset is not opened to public. The post-process data can be asked from the corresponding author by request, which cannot be used for commercial purpose.

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
