# Peer review of "Deep Learning-Based Auto-Segmentation of Spinal Cord Internal Structure of Diffusion Tensor Imaging in Cervical Spondylotic Myelopathy"

_diagnostics, 2023, doi:10.3390/diagnostics13050817_

Round 1

Reviewer 1 Report

manuscript showing the valuable potential of research.

Possible to publish after clarifying and supplementing some important issues.

1. The conclusions are too clear in terms of declaring positive verification of the CNNs used for automatic segmentation and determining the parameters of DTI metrics such as FA. This has not been proven.

Firstly:

DTI measurements as well as segmentations are for a single MR scanner. Thus, the results will be subject to systematic errors related to the scanner, the DTi sequence as well as its parameters. These systematic errors can be significant. Here I recommend the BSD-DTI problem and the Stejskal-Tanner equation for nonuniform gradients. Only after the elimination of systematic errors can we talk about a more universal material for teaching the CNN network. Now the results are very dependent on the hardware, the DTI sequence and its parameters. This should be duly discussed.

Secondly:

Aside from the first issue, we have a dependence on the scanner, the DTI sequence and its parameters: the size of the b matrix, the number of directions of the diffusion gradient vector, TE/TR times, resolution. DTI metrics depend on the parameters of the experiment. More experimental diversity is needed to be able to make such unequivocal judgments.

2. The literature could be supplemented with methodological, preclinical items on the Quantitative Assessment of Injury in Spinal Cords In Vivo by MRI.

3. The article should be supplemented with a vision of the research continuation path in order to verify the given hypotheses in a more precise way. Because, although there is no evidence in the article for the effectiveness of the algorithm (this applies only to this specific measurement, additionally burdened with systematic errors), it does not disqualify for the manuscript, because the way is potentially good, and the results obtained in a specific case can be generalized. But it is necessary to write what this research path should look like in the future and limit the conclusions in the manuscript to a specific case.

Author Response

  1. The conclusions are too clear in terms of declaring positive verification of the CNNs used for automatic segmentation and determining the parameters of DTI metrics such as FA. This has not been proven.

Reply:

Thanks for your advice, and the amendment has been made to the conclusion. (Page 13, lines 425-429)

Firstly:

DTI measurements as well as segmentations are for a single MR scanner. Thus, the results will be subject to systematic errors related to the scanner, the DTi sequence as well as its parameters. These systematic errors can be significant. Here I recommend the BSD-DTI problem and the Stejskal-Tanner equation for nonuniform gradients. Only after the elimination of systematic errors can we talk about a more universal material for teaching the CNN network. Now the results are very dependent on the hardware, the DTI sequence and its parameters. This should be duly discussed.

Secondly:

Aside from the first issue, we have a dependence on the scanner, the DTI sequence and its parameters: the size of the b matrix, the number of directions of the diffusion gradient vector, TE/TR times, resolution. DTI metrics depend on the parameters of the experiment. More experimental diversity is needed to be able to make such unequivocal judgments.

Reply:

Thanks and your advice are precious to us. We did the literature review about the factor that could affect the DTI measurement, and the supplement has been made in the discussion part. (Page 12, lines 405-413)

  1. The literature could be supplemented with methodological, preclinical items on the Quantitative Assessment of Injury in Spinal Cords In Vivo by MRI.

Reply:

The supplement has been made in the introduction section. (Page 1, lines 35-40)

  1. The article should be supplemented with a vision of the research continuation path in order to verify the given hypotheses in a more precise way. Because, although there is no evidence in the article for the effectiveness of the algorithm (this applies only to this specific measurement, additionally burdened with systematic errors), it does not disqualify for the manuscript, because the way is potentially good, and the results obtained in a specific case can be generalized. But it is necessary to write what this research path should look like in the future and limit the conclusions in the manuscript to a specific case.

Reply:

Thank you so much for your advice and recognition. We did have the continuation path for this research. The supplement has been made in the discussion section. (Page 12-13, lines 414-422)

Reviewer 2 Report

Comment #1: please include a few example images of the spinal cord with segmentation

Author Response

Thank you so much for your advice. The supplement has been made in section 3.2. Model performance. (Page 8, lines 261-264).

Language editing was made.

Reviewer 3 Report

Review of manuscript ”Deep learning-based auto-segmentation of spinal cord internal

structure of diffusion tensor imaging in cervical spondylotic myelopathy”. The authors have performed a study on deep learning-based auto-segmentation of the spinal cord in patients with cervical spondylotic myelopathy (CSM). They present a deep-learning model suitable for segmentation of the spinal cord. I have some comments from a clinical perspective.

Abstract

How many cases were the cervical slides from?

Material and Methods

Patients with clinical diagnosis of CSM were included. Was the CSM symptomatic in these patients?

Was the spinal medulla misshaped in some patients? The cervical disc herniation is usually causing pressure on the medulla causing it to lose its shape. How was this accounted for? I suppose by the manual drawing?

Discussion

Perhaps add a section regarding clinical applications, such as differ symptomatic from asymptomatic CSM? Degenerative changes in the cervical spine are widely common and issues regarding if these changes are symptomatic are commonly encountered.

Author Response

Abstract

How many cases were the cervical slides from?

Reply:

There were 89 cases in this research. The amendment has been made to the abstract (Page 1, line 18).

Material and Methods

Patients with clinical diagnosis of CSM were included. Was the CSM symptomatic in these patients?

Reply:

All of them are symptomatic. The Japanese Orthopaedic Association (JOA) Score is 9.78 ±3.48. (Page 3, lines 107 and 108)

Was the spinal medulla misshaped in some patients? The cervical disc herniation is usually causing pressure on the medulla causing it to lose its shape. How was this accounted for? I suppose by the manual drawing.

Reply:

Yes, there is the spinal medulla misshaped in some patients. This study aims to segment the spinal misshaped medulla by AI. The spinal misshaped medulla was segment manual as ground truth.

Discussion

Perhaps add a section regarding clinical applications, such as differ symptomatic from asymptomatic CSM? Degenerative changes in the cervical spine are widely common and issues regarding if these changes are symptomatic are commonly encountered

Reply:

Thank you so much for your advice. The amendment has been made in the discussion section. (Page 12, lines 382-397)

Language editing was made.

Round 2

Reviewer 1 Report

The authors supplemented the manuscript in the context of the impact of DTI parameters on the metrics, and also provided a certain vision regarding further research steps. However, they did not completely take into account the suggested issue regarding the impact of systematic errors on DTI metrics. This is the source of potentially large bugs. In a situation where we have one scanner, it is even more necessary to take into account. If we want to teach CNN, we strive to make the data as truthful as possible. Without eliminating the influence of systematic errors (especially related to spatially inhomogeneous magnetic field gradients), this does not seem to be possible. Therefore, I am asking you to include the thread of these errors in the DTI metrics and the future path of their elimination. Extensive information concerns the issue of BSD-DTI (B-matrix Spatial Distribution DTI) and the approach described by the Generalized ST Equation for nonuniform magnetic field gradients, which, as a special case, results in the commonly known Stejskal-Tanner equation, assuming the invariance of gradients in space, however, inconsistent with the experiment.

This is the last important issue, especially in the context of AI learning. Overall, the manuscript is much better now, but this point needs clarification.

Author Response

Thanks for your advice. We did the literature review in response to your advice. The amendment regarding the impact of systematic errors on DTI metrics has been made in the discussion part. (Page 13-14, lines 407-424).

Round 3

Reviewer 1 Report

I accept explanations. The article is now complete; taking into account the possibilities of current data.

On the subject of BSD-DTI, an article on "theoretical validation of the BSD-DTI method .." may (not necessarily) be added for the sake of literature order.